# Association of Antibiotic Use during the First 6 Months of Life with Body Mass of Children

**DOI:** 10.3390/antibiotics11040507

**Published:** 2022-04-11

**Authors:** Ji Hee Kwak, Seung Won Lee, Jung Eun Lee, Eun Kyo Ha, Hey-Sung Baek, Eun Lee, Ju Hee Kim, Man Yong Han

**Affiliations:** 1Department of Pediatrics, Kangbuk Samsung Hospital, School of Medicine, Sungkyunkwan University, Seoul 03181, Korea; hihikwak@gmail.com; 2Department of Data Science, College of Software Convergence, Sejong University, Seoul 05006, Korea; lsw2920@gmail.com (S.W.L.); mrt6519@gmail.com (J.E.L.); 3School of Medicine, Sungkyunkwan University, Suwon 16419, Korea; 4Department of Pediatrics, Kangnam Sacred Heart Hospital, Hallym University Medical Center, Seoul 07441, Korea; dmsry1@gmail.com; 5Department of Pediatrics, Kangdong Sacred Heart Hospital, Hallym University Medical Center, Seoul 05355, Korea; paviola7@gmail.com; 6Department of Pediatrics, Chonnam National University Hospital, Chonnam National University Medical School, Gwangju 61469, Korea; unelee99@gmail.com; 7Department of Pediatrics, CHA Bundang Medical Center, School of Medicine, CHA University, Seongnam 13496, Korea

**Keywords:** antibiotics, short stature, stunting, overweight, obesity, children

## Abstract

In this study, our objective was to assess the association of body mass in preschool children with the use of antibiotics within 6 months after birth. National administrative databases were used to examine all children born between 2008 and 2009 in Korea. Exposure was defined as the use of systemic antibiotics during the first 6 months of age. The observed outcomes were stunting (height for age [HFA] z score < −2.0), short stature (HFA z score < −1.64), overweight (body mass index [BMI] for age z score ≥ 1.04), and obesity (BMI for age z score ≥ 1.64), and the children’s height and body weight were measured from three to six years of age. To balance characteristics between the antibiotic user and non-user groups, propensity score matching was performed. The outcomes were evaluated using a generalized estimation equation with the logit link function. Analysis of antibiotic use by children during the first 6 months of life indicated there were 203,073 users (54.9%) and 166,505 non-users (45.1%). After PS matching, there were 72,983 antibiotic users and 72,983 non-users. Antibiotic use was significantly associated with stunting (aOR = 1.198, 95% CI = 1.056 to 1.360) and short stature (aOR = 1.043, 95% CI = 1.004 to 1.083), and had significant negative association with HFA z score (weighted β = −0.023). The use of an antibiotic for 14 days or more had a marked association with stunting. Antibiotic use was also associated with overweight, obesity, and increased BMI for age z score. Antibiotic use during the first 6 months of life increased the risk of stunting, short stature, overweight, and obesity in preschool children.

## 1. Introduction

Antibiotics have revolutionized the treatment of infectious diseases [1], but their excessive and inappropriate use has led to medical problems such as antibiotic resistance, adverse effects, and excessive medical costs [2,3]. In particular, the use of antibiotics during the prenatal and early postnatal periods can disrupt the establishment and maturation of healthy gut microbiota, thus contributing to health problems [4]. Several previous studies have reported that antibiotic exposure during early life was associated with obesity [5,6], allergic diseases [7,8,9], autoimmune diseases [10], celiac disease, and attention deficit hyperactivity disorder [11]. However, the rate of exposure to antibiotics in early life is quite high. A total of 50% of infants were exposed to at least one antibiotic in their first year of life in Australia [12]. In a cohort study of 1807 infants in the United States, approximately 40% of participants received at least one antibiotic treatment in their first year [13]. Unfortunately, Korea showed the highest antibiotic consumption rate among high-income countries [14].

Stunting and short stature are a serious public health problem and a major topic in children in developing and developed countries, respectively. Stunting is associated with dysfunction of the small intestine, a condition called pediatric environmental enteropathy. This condition is characterized by mucosal inflammation, villi flattening, and increased intestinal permeability [15]. An alteration of the normal intestinal commensal flora is a probable trigger of pediatric environmental enteropathy [4], and alteration of the normal intestinal flora may also drive the progression of stunting [4,16,17,18]. There is evidence of an association of the microbiota composition of meconium with height at ages of 1 and 2 years [16]. While systemic reviews and meta-analyses of the associations between antibiotic use in infancy and overweight/obesity have been reported [19], the understanding of the relationship between antibiotic use during early infancy and stunting or short stature in childhood is limited so far.

Our purpose is to evaluate the effect of antibiotic use within 6 months after birth with linear and ponderal growth in preschool children by analyzing an administrative nationwide cohort database for South Korea.

## 2. Methods

### 2.1. Study Design and Setting

This study was conducted using the data from the National Health Insurance Service (NHIS) of the Republic of Korea, a single-payer program that is mandatory for all residents. This retrospective population-based cohort used data from the 2008 National Investigation of Birth Cohort in Korea study (NICKs-2008), which consists of all 917,707 children who were born in Korea during 2008 and 2009 [20]. In brief, the NHIS maintains health records regarding healthcare utilization, prescriptions, and national health screening programs for the whole nation. The National Health Screening Program for Infants and Children (NHSPIC) is a population surveillance program that provides services to all health insurance subscribers, and consists of seven surveys taken from the age of 4 months to 72 months (first, age 4–6 months; second, age 9–12 months; third, age 18–24 months; fourth, age 30–36 months; fifth, age 42–48 months; sixth, age 54–60 months; seventh, age 66–72 months). The use of de-identified individual data for research purposes was authorized under the current National Health Insurance Act. The protocol of this study was reviewed and approved by the Institutional Review Board of the Korea National Institute for Bioethics Policy (P01-201603-21-005). The study followed recommended guidelines for observational studies that use routinely collected health data (Appendix A).

### 2.2. Data Sources

Patient demographic characteristics, health care utilization (type of visit, duration of hospitalization, medical costs, drug classification code, dose and duration of drug, disease information based on the *International Classification of Diseases 10th revision* [ICD-10]), and outcome data (height and weight) were from the database of the NICKs-2008 cohort. Body mass index (BMI) was calculated as weight in kilograms divided by height in meters squared. Health care utilization data were collected during the process of filing claims for healthcare services. Economic status was estimated by the amount of insurance co-payment and classified by quintiles. Residential status was classified as Seoul, metropolitan (Busan, Daegu, Incheon, Gwangju, Daejeon, and Ulsan), urban, or rural. The first round of the NHSPIC was implemented when subjects were 4 to 6 months old, the fourth round when they were 30 to 36 months old, the fifth round when they were 42 to 48 months old, the sixth round when they were 54 to 60 months old, and the seventh round when they were 66 to 71 months old. Physicians measured the height and weight of all children at each round. Appendix A provides additional information on the databases, variable definitions, and administrative codes.

### 2.3. Study Population

All included children had properly recorded birth weight information from the NHSPIC, completed the first round of the NHSPIC, and completed at least one round of the NHSPIC from rounds four to seven. Children were excluded if they were born before 37 weeks gestational age, had a birth weight less than 2.5 kg or more than 4 kg, or were admitted to an intensive care unit within 6 months after birth. A total of 369,578 children met the inclusion and exclusion criteria; 203,073 were prescribed antibiotics within 6 months after birth and were classified as antibiotic users, and 166,505 children did not use antibiotics within 6 months after birth and were classified as non-users. After 1:1 propensity score (PS) matching, 72,983 children were assigned to each group (Figure 1).

To assess the robustness of the results, an additional cohort consisting of users and non-users of antibiotics within 3 months after birth was also analyzed (Appendix A).

### 2.4. Antibiotic Use

Systemic use (oral, intramuscular, or intravenous) of an antibiotic during the first 6 months of age was examined, as this is considered the period when microbial colonization is susceptible to environmental factors [21,22]. Antibiotic use was identified in the records from the NHIS by drug classification codes for the prescription of a systemic antibacterial agent. In addition, antibiotic use was stratified into oral and parenteral (intramuscular or intravenous) according to the route of administration. To assess the duration-dependent effect of antibiotics, duration of use was classified by prescription days to differentiate 7 days or less vs. 8 days or more, and 13 days or less vs. 14 days or more. In matched data, about two-thirds of participants were exposed to antibiotics for 7 days or less at cumulative days. 21% of participants were exposed to antibiotic for 8 to 13 days, and only 17% were exposed to antibiotics for more than 13 days. Antibiotic use was analyzed as a continuous variable, a binary variable (use vs. none), and a multilevel categorical variable.

### 2.5. Outcomes

Anthropometric data, including height and weight, were measured at the fourth to seventh rounds of the NHSPIC (from 30 to 72 months old). This age was an appropriate period for follow-up, as most childhood obesity develops by the age of 5 years [23].

The primary outcome was stunting and short stature. Stunting was defined as the height for age [HFA] z score less than −2.0, and short stature was defined as the HFA z score less than −1.64 [24].

The secondary outcome was overweight and obesity. Overweight was defined as BMI for age z score of 1.04 or over, and obesity was defined as BMI for age z score of 1.64 or over [25].

### 2.6. Covariates

PS matching used the covariates in Appendix A to balance the antibiotic users and non-users in the main cohort. The demographic variables were age, sex, income quintile (based on insurance premium), and residence at birth. Nutrition, including breastfeeding during early infancy and additional complementary feeding before the first 3 months of age, were assessed in the first round of the NHSPIC questionnaire. Perinatal conditions and congenital/chromosomal abnormalities were evaluated by the ICD-10 codes from the NHIS. To identify clinical conditions within 6 months after birth, the most prevalent of 46 diseases, including respiratory infections, gastrointestinal infections, and dermatitis, were determined from ICD-10 codes. Use of non-antibiotic drugs, including those commonly used by infants and those that may alter growth during childhood, were determined from drug classification codes. Hospitalization records and emergency room visits within 6 months after birth were from the NHIS database.

The additional cohort (antibiotic user vs. non-user within 3 months after birth) was assessed using the same demographic characteristics, nutrition, and perinatal and congenital/chromosomal conditions. Clinical conditions were evaluated by prevalent disease history (ICD-10 codes), drug use (drug classification codes), and the number of hospitalizations and emergency room visits within 3 months after birth.

### 2.7. Statistical Analysis

PS matching was performed to reduce potential confounding and to balance the baseline characteristics between the antibiotic user and non-user groups by using the propensity score of the predicted probability of individuals with antibiotic exposure versus those without antibiotic exposure using a multivariable logistic regression model with adjustment for 79 *a priori* covariates (Appendix A). We matched both groups in a 1:1 ratio using a Mahalanobis algorithm with calipers of 0.01. Between-group differences in baseline characteristics were compared using standardized differences in unmatched and matched samples, and differences greater than 10% were considered meaningful [26].

For analysis of the associations of binomial variables of stunting, short stature, overweight, and obesity measured 4 times from 30 to 72 months of age with antibiotic use in early infancy, odds ratios (ORs) with 95% confidence intervals (CIs) were estimated using generalized estimated equations (GEE) with the logit link function. Adjusted β values and 95% CIs were estimated using mixed-model analysis to analyze the association of antibiotic use with HFA z score and BMI for age z score [27]. These analyses were adjusted for birth weight, sex, and breastfeeding status within 4 to 6 months after birth, residence at birth, and economic status.

Two pre-specified subgroup analyses examined types of feeding (breast feeding vs. no breast feeding) and birth weight (median value of birth weight = 3.24 kg, <3.24 kg vs. ≥3.24 kg). In addition, we examined the effect of antibiotic use on HFA z score and BMI for age z score by route of antibiotic administration (parenteral [intravenous or intramuscular] vs. oral).

A two-tailed P value less than 0.05 was considered significant. All statistical analyses were performed using SAS version 9.4 (SAS Institute Inc., Cary, NC, USA).

## 3. Results

### 3.1. Sociodemographic and Clinical Characteristics of Children

We initially analyzed the basic sociodemographic characteristics of children who were users and non-users of antibiotics during the first 6 months of life (Table 1). Before PS matching, all standardized differences were less than 10% except sex (13.9%) and residence at birth (10.3%). After PS matching, the two groups were balanced in all examined variables.

We also analyzed the basic clinical characteristics of children who were users and non-users of antibiotics (Table 2). Before PS matching, there were imbalances between the two groups in the number of hospitalizations (48.6%), the number of emergency room visits (17.4%), in respiratory and cardiovascular disorders (11.8%), and infectious diseases during the perinatal period (15.8%). Analysis of the most prevalent diseases and the use of major drugs within 6 months after birth indicated that most of the standardized differences were more than 10%. The number of children with malnutrition before matching was very small (9 vs. 10) and not significantly different between the groups before and after PS matching. After PS matching, there were no major imbalances in all examined clinical characteristics (Appendix A).

Our analysis of the basic sociodemographic and clinical characteristics of children in the additional cohort (antibiotic use vs. non-use within 3 months after birth) were also balanced after PS matching (Appendix A).

### 3.2. Main Outcome: Effect of Antibiotic Use on Stunting, Short Stature, and Linear Growth Faltering

We initially examined the effect of antibiotic use on HFA z score (Table 3 and Figure 2). In the matched data of the main cohort, more antibiotic users than non-users had stunting at preschool age (615 [0.84%] vs. 543 [0.74%]; aOR = 1.198, 95% CI = 1.056 to 1.360) and short stature at preschool age (1579 [2.16%] vs. 1489 [2.04%]; aOR = 1.043, 95% CI = 1.004 to 1.083). In addition, when we treated HFA z score as a continuous variable, antibiotic use had a negative association with HFA z score (aβ = −0.023, 95% CI = −0.031 to −0.015).

In contrast, analysis of the additional cohort indicated that antibiotic use was not significantly associated with stunting (500 [0.84%] vs. 1339 [2.25%]; aOR = 1.052, 95% CI = 0.918 to 1.205) or short stature (474 [0.79%] vs. 1311 [2.21%]; aOR = 1.073, 95% CI = 0.990 to 1.164) (Appendix A and Figure 2). As in the main cohort, when we treated HFA z score as a continuous variable, antibiotic use was negatively associated with HFA z score (aβ = −0.024, 95% CI = −0.033 to −0.015).

### 3.3. Effect of Duration of Antibiotic Use on Linear Growth Faltering

We then analyzed the effect of duration of antibiotic use on stunting and short stature at preschool age (Table 4 and Figure 2). The matched data indicated 543 children (0.74%) were non-users, 382 (0.83%) used an antibiotic for 7 days or less, 233 (0.86%) used an antibiotic for 8 days or more, and 108 (0.90%) used an antibiotic for 14 days or more. Relative to non-users, the risk of stunting was significantly greater for those who used antibiotics for 8 days or more (OR = 1.231, 95% CI = 1.040 to 1.456) and 14 days or more (OR = 1.299, 95% CI = 1.037 to 1.628). In addition, the risk of short stature was significantly associated with antibiotic use of 14 days or more (OR = 1.165, 95% CI = 1.013 to 1.339).

Further analysis indicated that the mean HFA z score decreased as the duration of antibiotic use increased in the main cohort and in the additional cohort (Figure 3A,B).

### 3.4. Secondary Outcome: Effect of Antibiotic Use on Overweight, Obesity, and Ponderal Growth

We next examined the effect of antibiotic use within 6 months after birth on ponderal growth (Table 3 and Figure 2). Among antibiotic users, 18,543 (25.41%) were overweight and 7044 (9.65%) were obese; among non-users, 18,287 (25.06%) were overweight and 6707 (9.19%) were obese. Children who used antibiotics had a significantly increased risk of overweight (aOR = 1.032, 95% CI = 1.007 to 1.056) and obesity (aOR = 1.064, 95% CI = 1.026 to 1.105). Increased BMI z score was also associated with antibiotic use (aβ = 0.015, 95% CI = 0.005 to 0.026).

In agreement, analysis of the additional cohort (Appendix A and Figure 2) indicated that antibiotic use within 3 months after birth was a significant risk factor for overweight (aOR = 1.029, 95% CI = 1.003 to 1.056) and obesity (aOR = 1.063, 95% CI = 1.021 to 1.107). Antibiotic use was also positively associated with BMI for age z score (aβ = 0.016, 95% CI = 0.005 to 0.028).

### 3.5. Effect of Duration of Antibiotic Use on Ponderal Growth

Examination of the effect of duration of antibiotic use on ponderal growth indicated that use for 8 days or more and 14 days or more significantly increased the risk for overweight and obesity (Table 5 and Figure 2). Further analysis indicated that BMI z score increased as the duration of antibiotic use increased in the main cohort and in the additional cohort (Figure 3C,D).

### 3.6. Prespecified Sensitivity and Subgroup Analysis

Our prespecified sensitivity and subgroup analysis showed that the relationship between antibiotic use and HFA z score was similar in children who were and were not breastfed during the first 4 to 6 months of age (Figure 4 and Appendix A). Antibiotic use had similar negative associations with HFA z in children who were breastfed (β = −0.026, 95% CI = −0.037 to −0.014) and those who were not breastfed (β = −0.020, 95% CI = −0.031 to −0.009). However, antibiotic use had positive associations with BMI for age z score in children who were breastfed (β = 0.015, 95% CI = 0.001 to 0.003), and had no significant associations in children who were not breastfed (β = 0.014, 95% CI = −0.001 to 0.029).

We performed a subgroup analysis by dividing children into upper-half and lower-half groups based on the median birth weight of 3.24 kg (Figure 4 and Appendix A). In this analysis, antibiotic use had significantly negative associations with HFA z score in both groups (lower-half: β = −0.029, 95% CI = −0.040 to −0.018; upper-half: β = −0.013, 95 CI = −0.024 to −0.001). Analysis of the additional cohort indicated that early antibiotic use and HFA z score were significantly associated in the upper-half subgroup. In addition, antibiotic use was positively associated with BMI for age z score in the lower-half and upper-half birthweight subgroups (lower-half: β = 0.016, 95% CI = 0.002 to 0.031; upper-half: β = 0.016, 95 CI = 0.001 to 0.032).

### 3.7. Routes of Antibiotic Administration

We analyzed the effects of antibiotic use on HFA z score and BMI for age z score according to the route of administration (Figure 4 and Appendix A). Parenteral administration had no significant association with HFA z score and BMI for age z score. However, oral administration had a significant negative association with HFA z score (β = −0.0178, 95% CI = −0.0283 to −0.0072) and a significant positive association with BMI for age z score (β = 0.019, 95% CI = 0.006 to 0.031).

## 4. Discussion

Our analysis of a large national administrative cohort showed that antibiotic use within the first 6 months after birth increased the risk for stunting and short stature in preschool children. The effect of antibiotic use on growth was especially evident in those who used antibiotics for longer durations. The significantly adverse effect of antibiotic use on linear growth faltering remained significant regardless of breastfeeding status within 4 to 6 months after birth and regardless of birth weight. In addition, antibiotic use during early infancy had a positive association with ponderal growth of preschool children.

Stunting is associated with changes in the microbiota of the small intestine, whose function is essential for digestion and absorption of nutrients [28]. Certain microbes in the small intestine, whose identities remain unknown, contribute to flattening of the villous, and eventually to childhood stunting [18,29,30]. Antibiotic use can induce microbiota alterations in the distal gut [31]. In addition, perturbation of the microbiota in the small intestine of experimental animals, which was induced by early-life antibiotic intervention, decreased the abundance of some *Lactobacillus* species and increased the abundance of some *Streptococcus* species [32,33]. However, there is a paucity of data regarding the association between stunting in children and changes in their intestinal microbiota due to antibiotic use. The present nationwide administrative cohort study showed that early use of antibiotics increased the risk for stunting, short stature, and linear growth faltering in preschool children. This result supports previous studies, which suggested that a higher abundance of *Staphylococcus* species in the microbiota of the meconium was negatively associated with height at ages 1 and 2 years [17]. To the best of our knowledge, the present study is the first to document a significant association between early antibiotic use and linear growth faltering in preschool children.

We assessed the robustness of our results by using two subgroup analyses, one based on breastfeeding status and the other based on birth weight. There is evidence that breastfeeding may prevent infections [34], overweight [35], and sudden infant death syndrome [36]. In addition, birth weight is a strong predictor of weight and height during early childhood among children with normal and low birth weight [37,38]. Our results indicated that antibiotic use during early infancy was associated with linear growth faltering in preschool children regardless of breastfeeding status or birth weight.

There is conflicting evidence regarding the effect of the route of antibiotic administration on the intestinal microbiota [39]. Study of an animal model indicated that oral and parenteral antibiotics induced dysbiosis, but oral antibiotics had a 100-fold stronger effect in reducing microbiota colonization [40]. Another animal study reported that oral and parenteral antibiotics induced significant dysbiosis of fecal microbiota [41]. Our findings suggest that oral antibiotic use was associated with linear growth faltering in preschool children, but parenteral antibiotic use was not. However, this result should be interpreted with caution due to the small number of children in our overall cohort who received parenteral antibiotics (15.8%).

Two previous meta-analyses reported associations between early antibiotic use and overweight and obesity during childhood [19,42]. Other researchers suggested that changes of the gut microbiota explained the effects of antibiotics on childhood obesity [43,44]. In support of these previous results, our results confirmed that antibiotic use during early infancy was a risk factor for overweight and obesity and was positively associated with ponderal growth in preschool children.

Previous research indicated that long-term use of an antibiotic had a more evident effect on ponderal growth [44,45]. In particular, Park et al. showed that relative to short-term users (cumulative days of use = 1–30), long-term users (cumulative days of use > 180) had a greater risk of obesity within the first 24 months of age (OR = 1.40, 95% CI = 1.19 to 1.64) [44]. Dawson-Hahn et al. showed that antibiotic use within the first 12 months of age increased the risk of obesity [13].

This study included a large set of data from many children, allowing the assessment of anthropometric indices measured by physicians in hospitals. Because all children born during 2008 and 2009 in Korea were enrolled and most of them participated in the NHSPIC, the data in this study have representative characteristics, and the results can be generalized. By using PS score matching, we tried to quantitatively assess the association between antibiotic use and childhood growth with adjustment for potential confounders, especially economic status and co-morbidities. In addition, in Korea, the rates of exposure to antibiotics in infants aged <3 months and <2 years old were high, approximately 25% and 99%, respectively [44]; by contrast, the exposure rate to antibiotics within the first 6 months of life in the United States is only about 14% [6]. Because Korea is a developed country with national health insurance and easy access to medical care, only a small number of children suffer from undernutrition. Our administrative data help to determine the impact of excessive and inappropriate antibiotic use at an early age on children’s health in a developed country.

However, the present study had some limitations. Firstly, though we used PS matching, the retrospective study design prevents us from inferring causal relationships. In particular, there is a possibility that children who used antibiotics, especially for long durations, had serious clinical conditions that contributed to growth faltering. Secondly, participants in this study were born in 2008 and 2009 and were followed up to their age of 6 years. Since time has passed, it is necessary to confirm the consistency of results with the latest data. Thirdly, we used antibiotic prescriptions to define antibiotic use, although we cannot be certain that prescribed antibiotics were actually administered. Fourthly, we did not have data regarding the different types of antibiotics. Fifthly, we were unable to assess some factors that might have affected the gut microbiota and child growth, including birth height, mode of birth delivery, maternal health status, parental height and weight, and secondary smoking status. Finally, multiple genetic and environmental factors that we could not assess could have affected growth. Therefore, a longitudinal prospective study is needed to confirm our results.

## 5. Conclusions

Our results suggest that antibiotic use during early infancy is a potential risk factor for stunting, short stature, overweight, and obesity during childhood. We suggest that clinicians more carefully consider the administration of antibiotics during early infancy because of these risks.

## Figures and Tables

**Figure 1 antibiotics-11-00507-f001:**
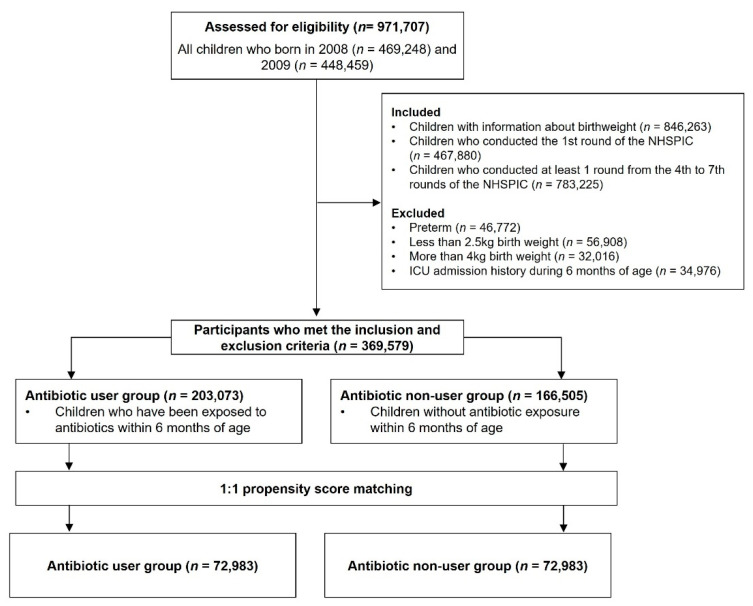
Enrollment, assessment of eligibility, and PS matching of antibiotic users and non-users.

**Figure 2 antibiotics-11-00507-f002:**
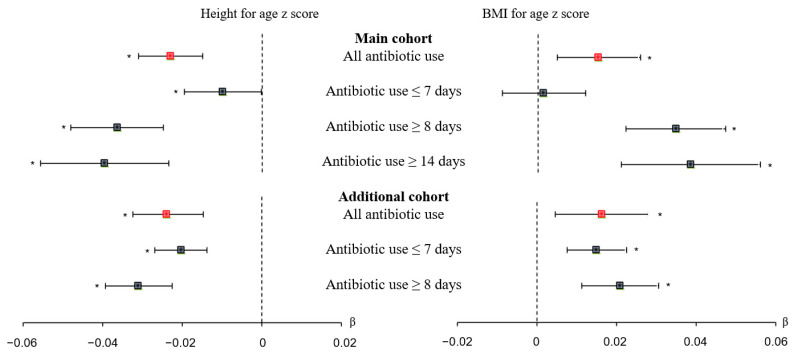
Effect of antibiotic use and duration of antibiotic use on linear growth (top) and ponderal growth (bottom) in the main cohort and in the additional cohort (forest plot). The main cohort and additional cohort consisted of participants who were antibiotic users or non-users within 6 months after birth and within 3 months after birth, respectively. Black filled rectangles indicate β, black lines indicate 95% CIs, and asterisks indicate *p* < 0.05.

**Figure 3 antibiotics-11-00507-f003:**
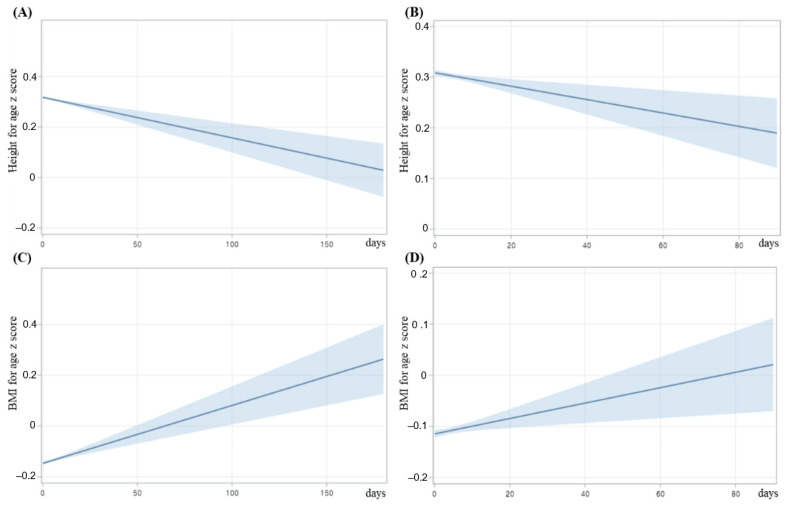
Effect of duration of antibiotic use on HFA z score (**A**,**B**) and BMI for age z score (**C**,**D**) in the main cohort (**A**,**C**) and in the additional cohort (**B**,**D**). Solid blue lines indicate means and light blue areas indicate 95% CIs.

**Figure 4 antibiotics-11-00507-f004:**
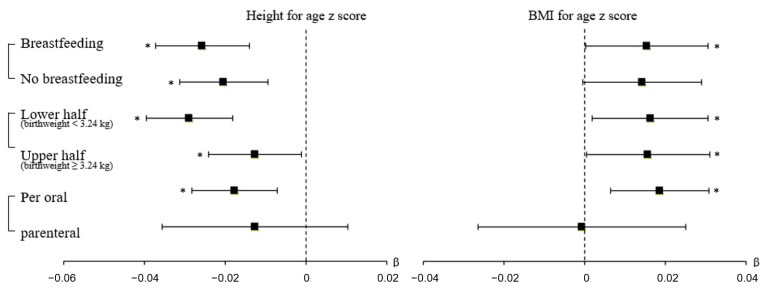
The effect of antibiotic use on HFA z score and BMI for age z score in different subgroups of the main cohort (forest plot) defined by breastfeeding status, birth weight, and route of antibiotic administration. Black filled rectangles indicate β, black lines indicate 95% CIs, and asterisks indicate *p* < 0.05.

**Table 1 antibiotics-11-00507-t001:** Basic sociodemographic characteristics of children in the main cohort ^a^.

Sociodemographic Characteristic	All Data (*n* = 369,578)			PS-Matched Data (*n* = 145,966) ^b^		
	Antibiotic, *n* (%) ^c^		Standardized Difference% ^d^	Antibiotic, *n* (%) ^c^		Standardized Difference % ^d^
	Users	Non-Users		Users	Non-Users	
	(*n* = 203,073)	(*n* = 166,505)		(*n* = 72,983)	(*n* = 72,983)	
Sex						
Female	92,603 (45.6)	87,366 (52.5)	13.9	36,194 (49.6)	36,309 (49.7)	0.3
Male	110,470 (54.4)	79,139 (47.5)		36,789 (50.4)	36,674 (50.3)	
Residence at birth ^e^						
Seoul	45,397 (22.4)	45,676 (27.4)	10.3	18,560 (25.4)	19,336 (26.5)	0.1
Metropolitan	49,148 (24.2)	37,678 (22.6)		17,891 (24.5)	16,578 (22.7)	
Urban	82,260 (40.5)	63,637 (38.2)		28,555 (39.1)	28,769 (39.4)	
Rural	24,582 (12.1)	17,915 (10.8)		7977 (10.9)	8300 (11.4)	
Birth year						
2008	101,262 (49.9)	76,782 (46.1)	7.4	35,387 (48.5)	35,368 (48.5)	0.1
2009	101,811 (50.1)	89,723 (53.9)		37,596 (51.5)	37,615 (51.5)	
Birth weight, kg (SD) ^f^	3.24 (0.32)	3.23 (0.32)	5.7	3.23 (0.32)	3.23 (0.32)	0.1
Type of feeding ^g^						
Only breastfeeding	90,246 (44.4)	77,897 (46.8)	1.3	33,700 (46.2)	34,139 (46.8)	0.1
Only formula milk	73,351 (36.1)	54,268 (32.6)		25,207 (34.5)	24,302 (33.3)	
Mixed	37,702 (18.6)	33,128 (19.9)		13,812 (18.9)	14,256 (19.5)	
Special milk	885 (0.4)	522 (0.3)		264 (0.4)	286 (0.4)	
Additional complementary feeding before 3 months of age ^h^						
Yes	88,625 (43.6)	68,927 (41.4)	4.6	31,344 (45.9)	31,438 (43.1)	0.3
No	112,992 (55.6)	96,530 (58.0)		41,639 (57.1)	41,545 (56.9)	
Economic status ^i^						
1 (Lowest)	16,105 (7.9)	11,894 (7.1)	6.9	5579 (7.6)	5579 (7.6)	0.8
2	30,567 (15.1)	23,319 (14.0)		10,827 (14.8)	11,044 (15.1)	
3 (Middle)	55,256 (27.2)	43,633 (26.2)		20,199 (27.7)	20,349 (27.9)	
4	63,483 (31.3)	54,222 (32.6)		24,393 (33.4)	24,090 (33.0)	
5 (Highest)	30,276 (14.9)	27,967 (16.8)		11,985 (16.4)	11,921 (16.3)	

Abbreviations: *n*, number; SD, standard deviation. ^a^ Unless otherwise specified, baseline characteristics were assessed on the date of birth. ^b^ Propensity score matching was performed to reduce bias for selection of the comparison group. Matching was performed using the Mahalanobis algorithm with a caliper of 0.01 and multivariable logistic regression with 79 previously chosen covariates (Appendix A). ^c^ Results are reported as N (%) unless otherwise indicated. ^d^ Differences greater than 10% were interpreted as meaningful. All standardized differences of cohort values were less than 0.05. ^e^ Metropolitan areas were defined as six metropolitan cities (Busan, Incheon, Gwangju, Daegu, Daejeon, and Ulsan), urban areas (cities), and rural areas (non-city areas). Of all participants, information was missing for 1686 users and 1599 non-users. ^f^ Obtained by the 1st NHSPIC at 4 to 6 months after birth. ^g^ Obtained by the 1st NHSPIC at 4 to 6 months after birth. Of all participants, information was missing for 889 users and 690 non-users. ^h^ Obtained by the 1st NHSPIC at 4 to 6 months after birth. Of all participants, information was missing for 1456 users and 1048 non-users. ^i^ Economic status was estimated by the amount of insurance co-payment and classified by quintiles. Of all participants, information was missing for 7386 users and 5470 non-users.

**Table 2 antibiotics-11-00507-t002:** Basic clinical characteristics of children in the main cohort ^a^.

Clinical Characteristic	All Data (*n* = 369,578)			PS-Matched Data (*n* = 145,966) ^b^		
	Antibiotic, *n* (%) ^c^		Standardized Difference % ^d^	Antibiotic, *n* (%) ^c^		Standardized Difference % ^d^
	Users (*n* = 203,073)	Non-Users (*n* = 166,505)		Users (*n* = 72,983)	Non-users (*n* = 72,983)	
Hospitalization within 6 months after birth						
Hospitalization	50,256 (24.7)	12,045 (7.2)	48.6	9402 (12.9)	8790 (12.0)	1.1
ER visits	16,360 (8.1)	6399 (3.8)	17.4	3906 (5.4)	3883 (5.3)	0.0
Conditions (ICD-10 code) originating during the perinatal period, N (%) ^c^						
Fetus and newborn affected by maternal factors and by complications of pregnancy, labor, and delivery	4799 (2.4)	2396 (1.4)	6.8	1500 (2.1)	1492 (2.0)	0.1
Respiratory and cardiovascular disorders specific to the perinatal period	10,677 (5.3)	4936 (3.0)	11.8	3180 (4.4)	3108 (4.3)	0.5
Infections during the perinatal period	33,680 (16.6)	18,593 (11.2)	15.8	10,209 (14.0)	10,164 (13.9)	0.2
Hemorrhagic and hematological disorders of fetus and newborn	67,110 (33.0)	50,367 (30.2)	6.1	23,356 (32.0)	23,292 (31.9)	0.2
Transitory endocrine and metabolic disorders specific to fetus and newborn	6028 (3.0)	3580 (2.2)	5.4	1947 (2.7)	1876 (2.6)	0.6
Digestive system disorders of fetus and newborn	6473 (3.2)	4151 (2.5)	4.2	2153 (3.0)	2182 (3.0)	0.2
Congenital malformations, deformations, and chromosomal abnormalities	21,216 (10.4)	14,131 (8.5)	6.8	6956 (9.5)	7013 (9.6)	0.3
Prevalent diseases (ICD-10 code) diagnosed within 6 months after birth						
Other and unspecified gastroenteritis and colitis of infectious origin	17,544 (8.6)	8203 (4.9)	14.8	5134 (7.0)	5122 (7.0)	0.1
Gastroenteritis and colitis of unspecified origin	18,590 (9.2)	9019 (5.4)	14.5	5574 (7.6)	5584 (7.7)	0.1
Acute conjunctivitis	11,520 (5.7)	5234 (3.1)	12.2	3351 (4.6)	3375 (4.6)	0.2
Conjunctivitis, unspecified	11,828 (5.8)	5842 (3.5)	11.0	3566 (4.9)	3620 (5.0)	0.4
Acute suppurative otitis media	21,733 (10.7)	430 (0.3)	47.1	468 (0.6)	412 (0.6)	0.3
Acute nasopharyngitis	80,109 (39.4)	46,375 (27.9)	24.7	25,304 (34.7)	22,595 (31.0)	0.9
Acute sinusitis, unspecified	16,024 (7.9)	2419 (1.5)	30.8	2174 (3.0)	2130 (2.9)	0.3
Acute tonsillitis, unspecified	17,803 (8.8)	14,436 (8.7)	0.5	6504 (8.9)	6545 (9.0)	0.2
Acute upper respiratory infection, unspecified	8741 (4.3)	3646 (2.2)	11.9	2347 (3.2)	2375 (3.3)	0.2
Pneumonia, unspecified	12,012 (5.9)	399 (0.2)	33.3	424 (0.6)	380 (0.5)	0.4
Acute bronchitis	77,144 (38.0)	14,992 (9.0)	72.7	13,859 (19.0)	13,539 (18.6)	1.1
Acute bronchiolitis	58,490 (28.8)	8549 (5.1)	66.3	8356 (11.4)	7895 (10.8)	1.8
Noninfective gastroenteritis and colitis, unspecified	13,931 (6.9)	8224 (4.9)	8.1	4620 (6.3)	4697 (6.4)	0.4
Constipation	10,322 (5.1)	7752 (4.7)	2.1	3684 (5.0)	3782 (5.2)	0.6
Impetigo (any organism, any site)	4655 (2.3)	2352 (1.4)	6.5	1461 (2.0)	1438 (2.0)	0.2
Other atopic dermatitis	13,350 (6.6)	11,162 (6.7)	0.5	5122 (7.0)	5070 (6.9)	0.3
Atopic dermatitis	37,833 (18.6)	31,269 (18.8)	0.3	14,229 (19.5)	14,242 (19.5)	0.0
Urinary tract infection, site not specified	9203 (4.5)	1193 (0.7)	24.1	1075 (1.5)	1045 (1.4)	0.3
Fever, unspecified	15,068 (7.4)	7366 (4.4)	12.6	4215 (5.8)	4180 (5.7)	0.2
Malnutrition	9 (0.0)	10 (0.0)	0.0	4 (0.0)	6 (0.0)	0.0
Drug use (classification code) within 6 months after birth						
Antipyretic	123,846 (61.0)	47,328 (28.4)	69.1	32,519 (44.6)	32,669 (44.8)	0.4
Psycho/nervous system drug	26,818 (13.2)	8887 (5.3)	27.6	6175 (8.5)	6217 (8.5)	0.2
Antihistamine	156,558 (77.1)	60,913 (36.6)	89.6	43,719 (59.9)	44,398 (60.8)	2.1
Respiratory system drug	174,725 (86.0)	69,910 (42.0)	103.2	50,853 (69.7)	51,580 (70.7)	2.3
Digestive system drug	166,479 (82.0)	69,225 (41.6)	91.3	47,331 (64.9)	47,942 (65.7)	1.9
Hormone drug	25,110 (12.4)	2890 (1.7)	42.3	2641 (3.6)	2493 (3.4)	0.8
Steroid	24,929 (12.3)	2743 (1.6)	42.6	2571 (3.5)	2423 (3.3)	0.8

Abbreviations: N, Number; ICD, International Classification of Diseases; ER, Emergency room. ^a^ Unless otherwise specified, all of baseline characteristics were assessed at 6 months after birth. ^b^ Propensity score matching was performed to reduce bias for selection of the comparison group. Matching was performed using the Mahalanobis algorithm with a caliper of 0.01 and multivariable logistic regression with 79 previously chosen covariates (Appendix A). ^c^ Results are reported as N (%) unless otherwise indicated. ^d^ Differences greater than 10% were interpreted as meaningful. All standardized differences of cohort values were less than 0.05.

**Table 3 antibiotics-11-00507-t003:** Effect of antibiotic use on risk of linear growth and ponderal growth of preschool children in the main cohort ^a^.

	All Data (*n* = 369,578)			PS-Matched Data (*n* = 145,966) ^b^		
	Antibiotic, *n* (%)		aOR ^f^ (95% CI)	Antibiotic, *n* (%)		aOR ^f^ (95% CI)
	Users (*n* = 203,073)	Non-Users ^c^ (*n* = 166,505)		Users (*n* = 72,983)	Non-Users ^c^ (*n* = 72,983)	
Primary outcome: linear growth						
Stunting ^d^	1729 (0.85)	1251 (0.75)	**1.199 (1.108 to 1.299)**	615 (0.84)	543 (0.74)	**1.198 (1.056 to 1.360)**
Short stature ^e^	4513 (2.22)	3391 (2.04)	**1.127 (1.073 to 1.183)**	1579 (2.16)	1489 (2.04)	**1.043 (1.004 to 1.083)**
aβ (95% CI) ^g^	**−0.043 (−0.048 to −0.038)**			**−0.023 (−0.031 to −0.015)**		
Secondary outcome: ponderal growth						
Obesity ^h^	20,226 (9.96)	14,801 (8.89)	**1.167 (1.139 to 1.195)**	7044 (9.65)	6707 (9.19)	**1.064 (1.026 to 1.105)**
Overweight ^i^	53,458 (26.32)	40,754 (24.48)	**1.127 (1.110 to 1.143)**	18,543 (25.41)	18,287 (25.06)	**1.032 (1.007 to 1.056)**
aβ (95% CI) ^g^	**0.057 (0.050 to 0.515)**			**0.015 (0.005 to 0.026)**		

Abbreviations, N, Number; aOR, adjusted odds ratio; CI, confidence interval; aβ, adjusted β. ^a^ The main cohort consisted of participants who were antibiotic users or non-users within 6 months after birth. ^b^ Propensity score matching was performed to reduce bias for selection of the comparison group. Matching was performed using the Mahalanobis algorithm with a caliper of 0.01 and multivariable logistic regression with 79 previously chosen covariates (Appendix A). ^c^ Reference group. ^d^ Stunting was defined as a height for age z score less than −2.0.^17^. ^e^ Short stature was defined as a height for age z score less than −1.64.^17^. ^f^ Adjusted odds ratios were assessed using a generalized estimating equation with a binomial distribution logit link function and exchangeable working correlation structure, with adjustment for birth weight, sex, and breastfeeding within 4 to 6 months after birth, residence at birth, and economic status. ^g^ The adjusted estimates and 95% CIs of height for age z score and BMI for age z score were assessed using a mixed model of GENMOD, adjusting for birthweight, sex, and breastfeeding status within 4 to 6 months after birth, residence at birth, and economic status. ^h^ Overweight was defined as BMI for age z-score ≥ 1.03.^18^. ^i^ Obesity was defined as BMI for age z score ≥1.64.^18^ Bold values indicate *p* < 0.05.

**Table 4 antibiotics-11-00507-t004:** The duration-dependent effect of antibiotic use on stunting and short stature in children.

Duration of Prescribed Antibiotics		All Data (*n* = 369,578)		PS-Matched Data (*n* = 145,966) ^a^		
		Subjects, *n*	Events, *n* (%)	Subjects, *n*	Events, *n* (%)	OR (95% CI)
Stunting ^b^						
Referent	Non-users	166,505	1251 (0.75)	72,983	543 (0.74)	Ref
	Duration ≤ 7 days	83,529	678 (0.81)	45,955	382 (0.83)	**1.179 (1.021 to 1.361)**
	Duration ≥ 8 days	119,544	1051(0.88)	27,028	233 (0.86)	**1.231 (1.040 to 1.456)**
	Duration ≥ 14 days	76,143	688 (0.90)	12,013	108 (0.90)	**1.299 (1.037 to 1.628)**
Short stature ^c^						
Referent	Non-users	166,505	3391 (2.04)	72,983	1489 (2.04)	Ref
	Duration ≤ 7 days	83,529	1784 (2.14)	45,955	979 (2.13)	1.066 (0.976 to 1.164)
	Duration ≥ 8 days	119,544	2729 (2.28)	27,028	600 (2.22)	1.091 (0.983 to 1.211)
	Duration ≥ 14 days	76,143	1775 (2.33)	12,013	284 (2.36)	**1.165 (1.013 to 1.339)**

^a^ Propensity score matching (1:1) was performed to reduce bias for the selection of the comparison group. Matching was performed using the Mahalanobis algorithm with a caliper of 0.01 and multivariable logistic regression with 79 previously chosen covariates (Appendix A). ^b^ Stunting was defined as the height for age z score <−2.0.^17^. ^c^ Short stature was defined as the height for age z score <−1.64.^17^. Bold values indicate *p* < 0.05.

**Table 5 antibiotics-11-00507-t005:** The duration-dependent effect of antibiotic use on overweight and obesity in children.

Duration of Prescribed Antibiotics		All Data (*n* = 369,578)		PS-Matched Data (*n* = 145,966) ^a^		
		Subjects, *n*	Events, *n* (%)	Subjects, *n*	Events, *n* (%)	OR (95% CI)
Obesity ^b^						
Referent	Non-users	166,505	14.801 (8.89)	72,983	6707 (9.19)	Ref
	Duration ≤ 7 days	83,529	8046 (9.63)	45,955	4391 (9.55)	**1.051 (1.008 to 1.097)**
	Duration ≥ 8 days	119,544	12,180 (10.19)	27,028	2653 (9.82)	**1.096 (1.042 to 1.152)**
	Duration ≥ 14 days	76,143	7851 (10.31)	12,013	1212 (10.09)	**1.114 (1.040 to 1.194)**
Overweight ^c^						
Referent	Non-users	166,505	40,754 (24.48)	72,983	18,287 (25.06)	Ref
	Duration ≤ 7 days	83,529	21,441 (25.67)	45,955	11,563 (25.16)	1.018 (0.991 to 1.046)
	Duration ≥ 8 days	119,544	32,017 (26.78)	27,028	6980 (25.83)	**1.055 (1.022 to 1.089)**
	Duration ≥ 14 days	76,143	20,549 (26.99)	12,013	3164 (26.34)	**1.076 (1.029 to 1.124)**

Abbreviation, BMI, body mass index. ^a^ Propensity score matching was performed to reduce bias for selection of the comparison group. Matching was performed by Mahalanobis algorithm with a caliper of 0.01 using multivariable logistic regression with 79 previously chosen covariates (Appendix A). ^b^ Overweight was defined as BMI for age z score ≥1.03.^18^. ^c^ Obesity was defined as BMI for age z score ≥1.64.^18^. Bold values indicate *p* < 0.05.

## Data Availability

This study was based on the National Health Claims Database (NHIS2019-1-560) established by the NHIS of the Republic of Korea. Applications for using NHIS data are be reviewed by the Inquiry Committee of Research Support; if the application is approved, raw data is provided to the applicant for a fee. We cannot provide access to the data, analytic methods, and research materials to other researchers because of the intellectual property rights of this database that is owned by the National Health Insurance Corporation. However, investigators who wish to reproduce our results or replicate the procedure can be used in the database, which is open for research purposes (https://nhiss.nhis.or.kr/, accessed on 14 August 2021).

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
