# Peer review of "Association of Antibiotic Use during the First 6 Months of Life with Body Mass of Children"

_antibiotics, 2022, doi:10.3390/antibiotics11040507_

Round 1
Reviewer 1 Report
The present study evaluates the influences of antibiotic use during the first 6 months of life on anthropometric data (height, weight) of a children cohort born in Korea between 2008 and 2009.
The data was collected long time ago (more than 10 years), and I think it might be inappropriate for the current times. This could be one of the limitations of this study. Moreover, in my opinion, the lack of information on the influence of antibiotics used (INN), including their route of administration on the anthropometric data, is one of the biggest shortcoming of this study.
Please find my suggestions and comments below:
- Abstract: The authors should explain the meaning of PS
- Introduction: The authors should present more epidemiologic data (e.g., antimicrobial consumption, antimicrobial resistance, adverse reactions, stunting, obesity, including in children, etc.)
- Lines 68-69: “consists of seven surveys taken from the age of 4 months 68 to 72 months” – authors should detail this paragraph
- Paragraph “Outcomes” - Authors should present the criteria (values) for the inclusion of children in the overweight or obese groups; respectively in the short stature or stunting groups (e.g. BMI, height)
- Paragraph “Antibiotic use”- Could authors provide information on the antibiotics used and their route of administration?
- Lines 142-143: “... Mahalanobis algorithm with a caliper of 0.01 and multivariable logistic regression 142 with 79 a priori covariates” – authors should present more details
- Lines 149-150: “... generalized estimated equations (GEE) with the GENMOD procedure and the logit link function...” – authors should present more details
- Line 155: Is 3.24 the median? Authors should specify this.
- Lines 181-182: How was the income scale built? What are the values for each step of the scale?
- The limitations of the study should be completed.
Reviewer 2 Report
The manuscript by Ju Hee Kim et al. describes interesting data on the impact of antibiotic prescription on weight and height in the first half of life.
This manuscript describes very interesting results but the methods need to be further justified.
The authors should prefer the passive voice throughout their manuscript.
The flowchart is very difficult to follow and check the number of patients. Follow article writing recommendations (such as the CONSORT flowchart, even if it is not an RCT).
Authors should justify the threshold used for duration of antibiotic therapy (7 or 14), or birth weight (3.24)?
There are typos on the reference line 365.
italicize "et al."
94.1 patients?
Did the authors analyze their data by stratifying by birthplace? They have the data (Table 1) but do not conclude on it.
Round 2
Reviewer 1 Report
The manuscript has been modified according to my suggestions.
Reviewer 2 Report
The authors have efficiently enhanced the quality of their manuscript according to m previous comments.
Even if limited, the manuscript deserves now, for me, publication in Antibiotics.